# Effect of the trajectory of exertional breathlessness on symptom recall and anticipation: A randomized controlled trial

Viktor Elmberg[1,2☺]*, Magnus Ekström[2☺]

**1** Department of Clinical Physiology, Blekinge Hospital, Karlskrona, Sweden, **2** Department of Clinical Sciences Lund, Respiratory Medicine and Allergology, Faculty of Medicine, Lund University, Lund, Sweden

☺ These authors contributed equally to this work.
* viktore@gmail.com

**Data Availability Statement:** All relevant data are available at the Hardvard dataverse repository at https://doi.org/10.7910/DVN/QMQOCV.

**Funding:** VE was funded by an unrestricted grant from the Scientific Committee of Blekinge County

## Abstract

### Background

Breathlessness is a major cause of physical limitation. Recalled breathlessness intensity may differ from experienced intensity and be influenced by the intensity trajectory including the 'peak-end rule'. The primary aim was to test if adding two minutes of low intensity exercise at the end of an exercise test would change the recalled breathlessness. Secondary aims included to analyse the impact of the peak and end exertional breathlessness intensity on breathlessness recall.

### Methods

Randomized controlled trial of 92 adults referred for exercise testing who were randomized (1:1), at test end, to 2 minutes of additional low intensity exercise (intervention; n = 47) or stopping at peak exertion (control; n = 45). Experienced breathlessness during the test and recalled intensity (30 min after the test) was assessed using the Borg CR10 scale.

### Results

Participants were aged a mean 59 years; 61% men; 79% reported a mMRC $\geq$1. There was no between-group difference in recalled breathlessness intensity, 5.51 ([95% CI] 5.00 to 6.01) *vs.* 5.73 (5.27 to 6.20; p = 0.52) in controls, even though the intervention group had a significantly lower end breathlessness (mean difference 0.96; 0.24 to 1.67; p = 0.009). Recalled exertional breathlessness was most strongly related to peak breathlessness ($r^2$ = 0.43). When analyzed together, end breathlessness did not add any explanatory value above that of peak breathlessness.

### Conclusion

Adding an episode of two minutes of lower exercise and breathlessness intensity at the end of an exercise test did not affect symptom recall, which was most strongly related to peak breathlessness intensity.

Council. ME was supported by the Swedish Society for Medical Research.

### Trial registration

ClinicalTrials.gov (NCT03468205).

## Introduction

Breathlessness during exertion is a major limiting factor for the individual´s physical capacity and activity in severe diseases and across the community at large [1–3]. Increased exertional breathlessness is often associated with reduced physical activity, spiraling deconditioning and further worsening of breathlessness, worse quality of life and prognosis [4].

Patients' recall of breathlessness during recent activities and daily life is the basis for clinical evaluation and management. Recalled symptoms can differ substantially from the actually experienced symptoms [5–7]. The recall of symptom intensity is affected by several factors including the experienced peak intensity and the intensity at the end of the episode [5–7]. This 'Peak-end rule' has been reported in studies of pain [5,7], anxiety [8] and in breathlessness in daily life [9] and during exercise [10]. Evidence from pain suggest that the trajectory of the symptom affects the person's symptom recall and evaluation of the situation [5]. It was demonstrated in a randomized trial that adding 3 minutes with decreased pain at the end of a colonoscopy decreased the patients' recalled total pain during the procedure, improved their overall perception of the event and made them more willing to participate in similar future procedures [5]. However, studies of the impact of symptom trajectory on recall of breathlessness are lacking.

Pulmonary rehabilitation training is the first line treatment for exertional breathlessness and deconditioning in cardiorespiratory disease [11]. General physical activity is also important and is associated with reduced breathlessness in daily life and improved outcomes [12–14]. However, the amount of training and level of physical activity that a person is willing to participate in is influenced by that person's perception of his own capacity, which, in turn, depends on recall of previous breathlessness intensity during exertion [15]. The person's anticipated breathlessness intensity at a potential future physical task is likely a major determinant of his willingness to participate in training as well as of the level of physical activity in daily life. To improve the effectiveness of cardio-pulmonary rehabilitation training and the patients' health status, new approaches that may modify recalled exertional breathlessness intensity to optimize training and promote physical activity are needed [11].

The primary aim of this randomized controlled trial (RCT) was to test the hypothesis that adding a period of lower breathlessness intensity at the end of a standardized exercise test would modify the participant's recall of the level of breathlessness during the test. Secondary aims were to test the effect on recalled exertion during the test and anticipated future exertional breathlessness and exercise capacity. We also aimed to evaluate the association between recalled breathlessness and peak and end experienced breathlessness during the test.

## Materials and methods

### Design and population

This was a parallel group RCT conducted at the Department of Clinical Physiology, Blekinge Hospital, Sweden. Outpatients referred for standard exercise testing between March and December of 2018 were screened. The reason for referral were in most cases suspected chronic coronary syndrome. Other reasons for the test were suspected exercise induced arrythmias,

occupational health screenings and breathlessness. Exclusion criteria were signs of clinical or cardiovascular instability or other contraindication to exercise testing; and inability to read, write or understand Swedish sufficiently to participate. At the end of the exercise test (before randomization), we also excluded patients who had an exercise test duration of $\leq 3$ minutes or peak breathlessness of $\leq 3/10$ on a Borg category-ratio scale (CR10) [16].

## Ethical considerations

This trial was approved by the ethics committee of Lund University in Sweden (Dnr: 2017/310). All participants provided informed written consent. The trial was registered with ClinicalTrials.gov (NCT03468205) before recruitment of the first participant. The study is reported in accordance with the Consolidated Standards of Reporting Trials (CONSORT) guidelines for reporting RCTs [17].

## Eligibility and consent

Study information and a pre-test questionnaire was sent home to patients referred for standard cycle exercise testing, who had given preliminary consent by phone. The pre-test questionnaire was completed by the participant at home or at the Department of Clinical Physiology before the exercise test. Eligibility was confirmed and written informed consent obtained for all participants by the investigator before starting the exercise test.

## The exercise test and randomization

A standard incremental, symptom-limited cycle exercise test was performed according to current international [18,19] and Swedish guidelines [20]. Initial workload was usually 30W for women and 50W for men although higher values (up to 90W) could be used depending on the participant's expected exercise capacity. Incremental increases were also dependent on expected exercise capacity and were 10W, 15W or 20W for all participants, with 10W increments used for older and frail patients while younger and fit patients had 20W increments. The aim was to obtain an exercise test of about eight minutes.

At the end of the regular exercise test, with the participant still on the test cycle, a sealed opaque envelope was broken by the staff with a code that randomly allocated the participant in a 1:1 ratio to either an additional two minutes of lower intensity exercise testing at about 50% of the peak workload (intervention group) or no additional testing (control group). If the participant in the intervention group was unable to continue cycling even at the lower exercise level the workload was decreased to about 25% of the peak workload.

## Assessments

**Pre-test questionnaire.** A questionnaire was completed at home or prior to the exercise test, including data on smoking status and quantity of current/earlier smoking; breathlessness in daily life during the past two weeks on a 4-point Likert scale between 1 (none) and 4 (severe); impact of exertional breathlessness during the last two weeks using the modified Medical Research Council (mMRC) scale [21]; previously diagnosed disease(s) or surgical treatment(s) including myocardial infarction, angina pectoris, atrial fibrillation, congestive heart failure, valve disease, asthma, chronic obstructive pulmonary disease (COPD), diabetes, cancer and other.

**Exercise test.** Resting electrocardiogram (ECG) and blood pressure were registered immediately prior to the exercise test. Assessments during the exercise test and intervention period included: breathlessness intensity was self-reported by the participant before the test,

every 2 minutes during exercise, at peak exercise, at the end of the intervention period, and at the completion of the test on a Borg CR10 scale; self-reported intensity of exertion every 2 minutes and at the completion of the test on the Borg rating of perceived exertion (RPE) scale between 6 (none) and 20 (maximal) [16,22]. Standard measurements of exercise testing were taken according to clinical routine as follows: ECG, heart rate and work rate were registered continuously during the test; systolic blood pressure was measured every 2 minutes; oxygen saturation was registered as needed. Participants were asked to rate chest pain, if any, on a Borg CR10 scale every 2 minutes or more often when needed. At time of randomization, the allocated group and potential reason for not being randomized were registered.

**Post-test questionnaire.**   About thirty minutes after completion of the exercise test, the participants completed a questionnaire. Participants graded their general test performance on a 4-point Likert scale between 1 ("very bad") and 4 ("very good"), and their intensity of breathlessness during the test on a Borg CR10 scale using the question 'How breathless were you during the exercise test?'. Intensity of exertion during the exercise test was graded on a Borg RPE scale (question: 'How exhausting did you perceive the exercise test?'). Participants rated their anticipated future exertional breathlessness on a Borg CR10 scale using the question 'How breathless do you estimate that you would be if you were to do a similar exercise test in a few days' time'. Estimated intensity of exertion if a similar exercise test would be done in a few days, were rated on a Borg RPE scale (question: 'How exhausted do you estimate that you would be if you were to do a similar exercise test in a few days' time?'). Current self-perceived physical capacity was estimated on a numerical rating scale between 0 (worst imaginable) and 10 (best imaginable) (question: 'Estimate your physical capacity').

## Statistical analyses

Baseline patient characteristics was summarized using mean with standard deviation (SD) and median with range or interquartile range (IQR) for continuous variables with normal and skewed distribution, respectively. Categorical variables were expressed as frequencies and percentages. Differences between the groups were tested with t-tests for continuous and chi-square tests for categorical variables. Normal distribution was confirmed using histogram plots. Non-normal continuous variables were compared using Wilcoxon Rank-Sum test.

The primary outcome and main secondary outcomes were compared between groups using Student's t-test. The overall perception of the exercise test (4-point ordinal scale) was analyzed using Kruskal Wallis test. All comparisons were conducted by allocated group according to the 'intention-to-treat' principle and included all randomized participants.

The relations between recalled breathlessness and the peak and end breathlessness during exercise were analyzed using linear regression in the intervention group. The control group could not be included in this analysis as their breathlessness intensity at the peak and end of the test were the same (as they terminated the test at peak exertion).

Estimates were presented with 95% confidence intervals (CIs). Statistical significance was defined as two-sided p-value < 0.05. Statistical analyses were conducted using the software packages Stata, version 14.2 (StataCorp LP; College Station, TX).

## Sample size

An initial sample size of 74 randomized participants was pre-specified in the study protocol for 80% power to detect a clinically important between-group difference of 1 point on the Borg CR10 scale for the primary endpoint (recalled overall breathlessness) using Student's t-test, assuming a two-sided significance level (alpha) of 0.05 and a SD of 1.5 based on Stulbarg *et al.* [10] and Meek *et al.* [9] After 74 completed patients, an analysis blinded for group

allocation showed a slightly larger actual SD than that assumed, and the sample size was increased to at least 90 randomized patients to assure the pre-specified power for the primary endpoint.

## Results

### Participants

Between March and December 2018, 96 patients were included (Fig 1); four were excluded prior to randomization (two declined to participate and two exercised for ≤ three minutes). A total of 92 participants (61% men) were randomized to either the intervention group (n = 47) or the control group (n = 45). All participants completed the allocated intervention. Participant characteristics were similar between groups (Table 1). Participants had a mean age of 59 years, 15% were current smokers and 42% were previous smokers. 64% reported a modified Medical Research Council scale score (mMRC) of = 1 and 14% an mMRC ≥ 2.

### Exercise test and experienced breathlessness

Both groups exercised to similar peak breathlessness, mean difference 0.02 (95% CI, -0.60 to 0.65; p = 0.94). Peak level of exertion was also similar, mean difference 0.26 (95% CI, -0.34 to 0.85; p = 0.39). As shown in Table 2, the exercise time to randomization was somewhat longer in the control group (8.76 *vs*. 8.00 min) and mean peak load (W) was higher, mean difference 15.84 (95% CI, -6.81 to 38.5; p = 0.17). After randomization, the intervention group received two minutes of additional exercise (mean 2.07 min ± 0.19). The intervention group had a significantly lower last measured breathlessness during exercise, mean difference 0.96 (95% CI, 0.24 to 1.67; p = 0.009). Fig 2 shows the mean trajectory of breathlessness by study group.

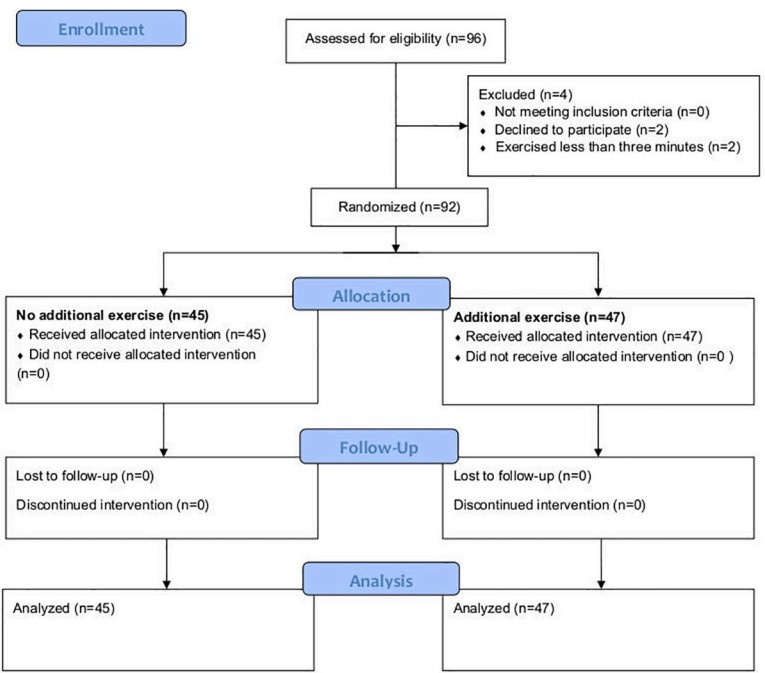

**Fig 1. CONSORT participant flow diagram.**

**Table 1. Patient baseline characteristics.**

| Characteristic | Intervention group | Control group |
|---|---|---|
| N | 47 | 45 |
| Age, years | 60.4 ± 14.2 | 57.6 ± 12.7 |
| Males | 31 (66%) | 25 (56%) |
| Smoking status | | |
| Current smokers | 8 (17%) | 7 (15%) |
| Former smokers | 18 (38%) | 21 (47%) |
| Never smokers | 21 (45%) | 17 (38%) |
| Mean years of smoking | 18.30 ± 19.33 | 17.77 ± 12.51 |
| Mean number of cigarettes per day in ever-smokers | 10.15 ± 5.58 | 11.80 ± 4.88 |
| Most common self-reported conditions | | |
| Hypertension | 9 (19%) | 10 (22%) |
| Hyperlipidemia | 4 (9%) | 7 (16%) |
| Diabetes | 2 (4%) | 6 (13%) |
| Asthma | 5 (11%) | 1 (2%) |
| Atrial fibrillation | 2 (4%) | 2 (4%) |
| Chronic obstructive pulmonary disease | 1 (2%) | 0 (0%) |
| mMRC breathlessness score | | |
| 0 | 10 (21%) | 10 (22%) |
| 1 | 31 (66%) | 28 (62%) |
| 2 | 5 (11%) | 6 (13%) |
| 3 | 1 (2%) | 1 (2%) |
| 4 | 0 (0%) | 0 (0%) |

Data presented as mean ± standard deviation, or frequency (%).

*Abbreviations*: mMRC = modified Medical Research Council scale.

## Primary and secondary endpoints

As shown in Table 3, there was no significant difference in the level of recalled mean breathlessness between the groups, mean difference 0.22 (95% CI, -0.45 to 0.9; p = 0.52). Anticipated

**Table 2. Values during exercise test per group.**

| Factor | Intervention | Control | P-value |
|---|---|---|---|
| N | 47 | 45 | |
| Exercise time to randomization, minutes* | 8.00 (1.96) | 8.76 (1.73) | 0.045 |
| Duration of additional exercise after randomization, minutes* | 2.07 (0.19) | 0 | <0.001 |
| Total duration of exercise, minutes* | 10.1 (1.98) | 8.8 (1.73) | 0.003 |
| Duration of recovery phase, minutes* | 10.2 (0.28) | 10.0 (1.09) | 0.94 |
| Peak workload, W | 156.6 (54.0) | 172.4 (55.5) | 0.17 |
| Peak breathlessness during exercise, Borg CR10 | 6.51 (1.60) | 6.53 (1.41) | 0.94 |
| Breathlessness at the end of exercise, Borg CR10 | 5.57 (1.90) | 6.53 (1.41) | 0.009 |
| Peak breathlessness during exercise and recovery, Borg CR10 | 6.53 (1.64) | 6.80 (1.53) | 0.42 |
| Peak perceived exertion during exercise, Borg RPE | 16.3 (1.35) | 16.5 (1.53) | 0.39 |
| Peak chest pain during exercise and recovery, Borg CR10 | 0.43 (1.04) | 0.39 (1.20) | 0.89 |

Data presented as means (standard deviation). Differences were evaluated using t-test. Durations (*) were compared using Wilcoxon Rank-Sum test. *Abbreviations*: Borg CR10 = Borg Category-ratio scale (0–10); Borg RPE = Borg rating of perceived exertion scale (6–20); W = Watt.

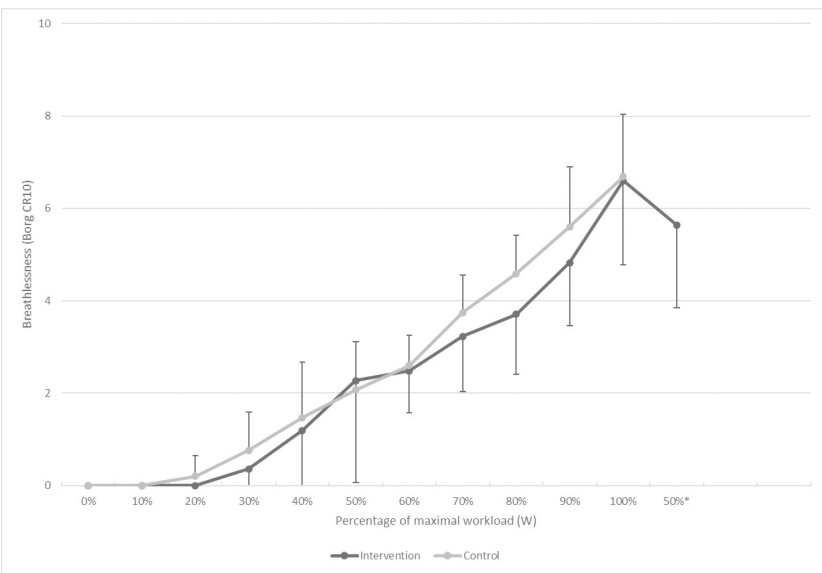

**Fig 2. Breathlessness during the exercise test by study group.** Mean trajectory of breathlessness by study group. Each point shows the mean breathlessness score (Borg CR10) related to percentage of maximal workload (W). The last point marked '50%*', shows the breathlessness score in the intervention group, at the end of the two minutes of low intensity exercise (about 50% of the maximal workload). Peak breathlessness was similar between the groups. End breathlessness was significantly lower in the intervention group compared to the control group, mean difference 0.96 (95% CI, 0.24 to 1.67). Error bars shows 1 SD. The first two SD for the intervention group was omitted from the graph for legibility (0.66 for 0.5 minutes and 1.05 for 1.5 minutes respectively). *Abbreviations*: SD, standard deviation; Borg CR10, Borg Category-ratio scale (0–10).

future breathlessness if a new exercise test were to be done in a few days' time were also without significant difference, mean difference 0.22 (95% CI, -0.49 to 1.46; p = 0.50). Recalled exertion and anticipated future exertion and current self-perceived physical capacity was without between-group difference (Table 3).

The data was complete regarding the primary endpoint. A small amount of data, in total 4 data points, was missing regarding secondary endpoints. The control group was missing one value regarding recalled exertion and one value regarding physical capacity. The intervention

**Table 3. Primary and secondary study endpoints.**

| Endpoints | Intervention N = 47 | Control N = 45 | Mean difference (95% CI) | P-value |
|---|---|---|---|---|
| **Primary:** | | | | |
| Recalled breathlessness, Borg CR10 | 5.51 ± 1.71 | 5.73 ± 1.56 | 0.22 (-0.45–0.90) | 0.52 |
| **Secondary:** | | | | |
| Recalled exertion, Borg RPE | 15.48 ± 1.74 | 15.40 ± 1.63 | -0.08 (-0.78–0.62) | 0.82 |
| Recalled performance (1–4) | 3.02 ± 0.50 | 3.02 ± 0.40 | 0.00 (-0.19–0.19) | 1.00 |
| Anticipated breathlessness if a new stress test were to be done in a few days' time, Borg CR10 | 4.53 ± 1.97 | 4.96 ± 1.76 | 0.42 (-0.35–1.20) | 0.28 |
| Anticipated exertion if a new stress test were to be done in a few days' time, Borg RPE | 14.42 ± 2.55 | 14.91 ± 2.12 | 0.49 (-0.49–1.46) | 0.32 |
| Physical capacity (0–10) | 5.13 ± 1.76 | 4.86 ± 2.00 | -0.27 (-1.06–0.52) | 0.50 |

A total of four data points was missing regarding secondary endpoints. In the control group one value regarding recalled exertion and one regarding physical capacity was missing. The intervention group lacked one value regarding recalled performance and one value regarding physical capacity. *Abbreviations*: Borg CR10 = Borg Category-ratio scale (0–10); Borg RPE = Borg rating of perceived exertion scale (6–20).

**Table 4. Associations for mean, peak and last breathlessness during exercise in the intervention group (n = 47).**

| Regression model: | Association with recalled breathlessness, linear regression Beta-coefficient (95% CI) and $r^2$ | |
|---|---|---|
| | Peak | Last |
| Each separate | 0.68 (0.44–0.92) $r^2$ = 0.43 | 0.41 (0.18–0.64) $r^2$ = 0.22 |
| Both (multivariable) | 0.75 (0.36-1-14) $r^2$ = 0.43 | -0.07 (-0.39–0.25) $r^2$ = 0.43 |

Associations with recalled breathlessness was analyzed in the intervention group who were randomized to an additional two minutes of exercise at a lower workload at the end of a regular exercise test. The control group was not included in the analysis as they stopped exercising at peak exertion with similar peak and last measured values of breathlessness. Associations with recall were estimated using linear regression for peak and end breathlessness separately (crude), and when analyzed together (each adjusted for the other); $r^2$ is the variance in the outcome (recalled breathlessness) explained by each model. *Abbreviations*: CI, confidence interval.

group was missing one value regarding recalled performance and one value regarding physical capacity. Due to the high completeness of data, analyses were by complete cases only. No data were imputed.

## Evaluation of the 'peak-end rule'

Recalled breathlessness intensity was associated with peak and end breathlessness during the exercise test when analyzed separately (Table 4). Peak breathlessness had a stronger relationship to recalled breathlessness ($r^2$ = 0.43, p < 0.001) than had end breathlessness ($r^2$ = 0.23; p = 0.001). Adding end breathlessness to peak did not increase the variance explained by the model above the variance explained by the peak value alone. Future anticipated breathlessness if a new stress test were to be performed were also more strongly related to peak than end breathlessness although the associations were weaker ($r^2$ = 0.16 vs. 0.11; p = 0.006 and 0.025 respectively). Again, end breathlessness did not increase the variance of recalled breathlessness explained above that for peak breathlessness.

## Sensitivity analysis

Mean peak load (W) and mean exercise time was somewhat higher in the control group compared to the intervention group (Table 2). In a post hoc analysis, this difference was related to a higher proportion of occupational health screening tests including firefighters in the control group, 9 out of a total of 11. The health screening subgroup had a significantly higher mean peak load (W) compared to participants who performed the test for non-screening reasons (233 *vs.* 154 W). When this group was excluded, mean peak load (W) was similar between the groups, mean difference 6.5 (95% CI, 15.48 to 28.49; p = 0.56). All findings where similar when excluding the health screening group; there was no difference in the in the level of recalled breathlessness between the groups, mean difference 0.08 (95% CI, -0.66 to 0.81; p = 0.83) or in any of the secondary outcomes.

## Discussion

We examined if adding a period of lower level exercise at the end of a standard exercise test would result in a lower level of recalled breathlessness and a lower level of anticipated future breathlessness. The main finding is that, even though the intervention group had a lower intensity of breathlessness at end exercise (mean difference 0.96; 95% CI, 0.24 to 1.67) which is likely to be clinically significant [23], we found no effect on the evaluated endpoints. This is to

our knowledge the first study to test the effect of the trajectory of exertional breathlessness on symptom recall and anticipation.

Our evaluation of the "peak-end rule" found that both peak and end breathlessness was related to recalled breathlessness, but peak was most strongly related. When analyzed together, peak remained strongly associated whereas end was not independently associated to the level of recalled breathlessness. This result contrasts with earlier studies of the "peak-end rule" that have found end intensity to be most important for recall of the evaluated sensation. These studies however differ from ours both regarding what was studied, and the methodology used. For example, Meek and colleagues studied recall of breathlessness during daily life over a long period while by Müller et al studied anxiety. Also, neither Meek nor Müller adjusted the end sensation intensity for peak intensity [8,24]. The finding that peak breathlessness and not end breathlessness predicted recall could in fact explain the study results. While we randomized the groups to different end breathlessness, peak breathlessness did not differ between the groups and a difference in recall is thus not to be expected. Peak breathlessness was marginally (non-significantly) higher immediately post exercise in the control group, which was likely caused by the buildup of oxygen debt with the participants likely having further increased ventilation after cessation of exercise.

A strength of the present study is the randomized controlled design and assessments during standardized exercise testing. Performance and procedures of the testing was likely to be similar for the groups as the randomization was performed at the end of the standard test. Up to that point, group allocation was unknown to both participants and staff. All participants received standardized procedures and information, and main endpoints were assessed using validated scales. Sample size was revised based on the actual SD (blinded to group allocation) to ensure adequate power for the primary endpoint.

A few potential limitations should be considered. The time between the end of the test (30 minutes) and completion of the post-test questionnaire might have been too short for adequate memory formation. The time the participants exercised at low intensity (2 minutes) might not have been enough to strongly influence the participants perception of the test. The intervention time was based on the Redelmeier study that evaluated a similar hypothesis in pain and used an intervention period of three minutes [5]. There is also a possible limitation in that the intervention group did not have a total exercise time that was 2 minutes longer than the control group. This was caused by a random difference with a larger proportion of participants, in the control group, who performed the test due to legally required occupational health screenings. This should however not be a major limitation because it resulted in a higher exercise level in the control group and therefore a larger between-group difference. Further, we analyzed the endpoints separately when excluding occupational health screenings which yielded similar findings for all outcomes. Symptom duration has also in studies of pain been shown to be of less importance compared to peak intensity, supporting the view that the difference in exercise time between the groups was not an important bias in the present study [5]. The findings may not be generalizable to patients with more severe physical limitation from breathlessness during daily life. Exercise testing in a clinical physiology laboratory provides highly standardized testing conditions but is performed in a secure environment and the breathlessness experienced might not be representative to that of activities in daily life.

The finding that recalled breathlessness and future anticipated breathlessness was more strongly related to peak than to end breathlessness could be potentially useful when tailoring cardiopulmonary exercise programs. It seems reasonable that cardiopulmonary exercise programs should strive to avoid extreme peaks of breathlessness to allow a gradual increase of exertional capacity. This is especially important for people with chronic breathlessness who are likely to experience physical limitation and anxiety due to their breathing problems. It is

also supported by American and European guidelines for pulmonary rehabilitation that recommends a target intensity of 12 to 14 on the Borg RPE scale (somewhat hard) for endurance training [25]. Lower peak breathlessness likely produces less adverse emotional responses potentially leading to an increased sense of physical capacity, self-sufficiency and improved physical activity and participation in rehabilitation training. Enrollment and completion of pulmonary rehabilitation training programs is still relatively low and further research is needed to validate the role of the symptom trajectory on recalled breathlessness.

Future studies should evaluate longer time periods and requirement of an even lower and more prolonged breathlessness value at the end of exercise, to optimize the between group difference. Future studies could also try to only include patients with breathlessness as their primary symptom. Our findings do not exclude the possibility that the trajectory of experienced breathlessness can affect memory formation and increase self-efficacy, participation in training and physical activity in daily life.

In conclusion, our result does not support the hypothesis that adding an episode of two minutes of low intensity exercise at the end of an exercise test affects the recall of an episode of exertional breathlessness. Peak breathlessness intensity was more strongly related to symptom recall than end breathlessness, which could have implications on the design of training interventions and cardiopulmonary rehabilitation.

## Supporting information

**S1 File. CONSORT 2010 checklist.**
(DOC)

**S2 File. Study protocol.**
(DOCX)

**S1 Fig. Breathlessness during the exercise test by study group.** Mean trajectory of breathlessness by study group. Each point shows the mean breathlessness score per minute (Borg CR10). Peak breathlessness was similar between the groups. End breathlessness was significantly lower in the intervention group compared to the control group, mean difference 0.96 (95% CI, 0.24 to 1.67). Error bars shows 1 SD. The first two SD for the intervention group was omitted from the graph for legibility (0.66 for 0.5 minutes and 1.05 for 1.5 minutes respectively). *Abbreviations*: SD, standard deviation; Borg CR10, Borg Category-ratio scale (0–10).
(TIF)

## Acknowledgments

The authors thank Professor Olle Pahlm, Lund University, Department of Clinical Physiology for invaluable advice, Helen Åsberg, Department of Clinical Physiology, Karlskrona for indispensable administrative assistance. We also thank Emma Lejon, Magdalena Klos-Piontek, Jenny Löfgren and Markus Holmgren who conducted the exercise tests, and all patients who participated and made this research possible.

## Author Contributions

**Conceptualization:** Viktor Elmberg, Magnus Ekström.

**Data curation:** Viktor Elmberg, Magnus Ekström.

**Formal analysis:** Viktor Elmberg, Magnus Ekström.

**Funding acquisition:** Viktor Elmberg, Magnus Ekström.

**Investigation:** Viktor Elmberg.

**Methodology:** Magnus Ekström.

**Project administration:** Viktor Elmberg.

**Resources:** Magnus Ekström.

**Software:** Magnus Ekström.

**Supervision:** Magnus Ekström.

**Writing – original draft:** Viktor Elmberg.

**Writing – review & editing:** Viktor Elmberg, Magnus Ekström.

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
