## [Decision Letter · Decision Letter 0]

11 Jun 2020

PONE-D-20-04503

Effect of the trajectory of exertional breathlessness on symptom recall and prediction: a randomized controlled trial

PLOS ONE

Dear Dr. Elmberg,

Thank you for submitting your manuscript to PLOS ONE. After careful consideration, we feel that it has merit but does not fully meet PLOS ONE’s publication criteria as it currently stands. Therefore, we invite you to submit a revised version of the manuscript that addresses the points raised during the review process.

We look forward to receiving your revised manuscript.

Kind regards,

Natasha McDonald

Associate Editor

PLOS ONE

Journal Requirements:

3. Please ensure that you refer to Figure 2 in your text as, if accepted, production will need this reference to link the reader to the figure.

Reviewers' comments:

Reviewer's Responses to Questions

**Comments to the Author**

1. Is the manuscript technically sound, and do the data support the conclusions?

Reviewer #1: Yes

Reviewer #2: Yes

Reviewer #3: Yes

2. Has the statistical analysis been performed appropriately and rigorously? 

Reviewer #1: Yes

Reviewer #2: Yes

Reviewer #3: Yes

3. Have the authors made all data underlying the findings in their manuscript fully available?

Reviewer #1: Yes

Reviewer #2: No

Reviewer #3: No

4. Is the manuscript presented in an intelligible fashion and written in standard English?

Reviewer #1: Yes

Reviewer #2: Yes

Reviewer #3: Yes

5. Review Comments to the Author

Reviewer #1: This paper tests the hypothesis that individuals who are randomly allocated to an additional 2 minutes of low intensity exercise following a standardised exercise test would experience lower recalled breathlessness than those who finished immediately upon completing their exercise test. The authors should firstly be commended for a very clear, concisely written manuscript. The study protocols and aims are also very explicit, and the null finding discussed in a thorough manner.

Minor comments:

- Patient population not described in abstract nor methods – this makes the reader curious as to why they were referred for exercise testing? There must have been other existing conditions that lead to this prescription for an exercise test? This is somewhat covered in the table of patient demographics, so it would be useful to have an in-text reference to this. A short overview in the methods section text would also be helpful.

- The question “How are your physical capacity” does not make grammatical sense in English – is there a more appropriate translation (I imagine this question was delivered in Swedish originally)?

- Statistical analyses: Were non-normally distributed variables compared using non-parametric tests? It is stated that they are described as a mean ± IQR, but were they also compared between groups in a non-parametric fashion instead of t-tests? It is not clear from the current description in the methods. It would also be helpful to have an additional column in the table of results that depicts which test (with the parametric or non-parametric nature) was employed for each variable.

- Table 4 is a little confusing. Could this be adjusted to two lines, as I understand the last two lines are part of the same regression analysis, with the column on the left depicting whether the estimates were calculated using separate or simultaneous regressions?

- Figure 2 would benefit from error bars on the graph, so that it is possible to visualise the variability in participant responses at each time point. A vertical line at the end of the incremental exercise might also be helpful to see when incremental exercise was terminated. Alternatively (or additionally in a second Figure panel), it could be helpful to see breathlessness scores plotted against exercise intensity - possibly as a percentage of maximum for consistency across participants.

- How were the missing values dealt with? Participants excluded, a correction method used? It would be helpful if this was described.

- It might also be useful to plot and test the correlation between peak and observed breathlessness, to see if this is the reason for no additional variance being explained when both regressors are included in the analysis together.

Reviewer #2: Effect of the trajectory of exertional breathlessness on symptom recall and prediction: a

randomized controlled trial

Viktor Elmberg.

Manuscript Number: PONE-D-20-04503

Summary

The authors compared ratings for dyspnea (Borg CR-10) and perceived exertion (Borg 6-20) among a sizeable number of adults referred for CPX. Subjects were randomized into control and intervention groups: latter pedaled an additional 2 min. at 25%-50% peak work (most at 50%) and were asked at the end of this cool-down to again rate dyspnea and perceived exertion. Authors wished to test hypothesis that cool-down would modify participant’s recall of the level of breathlessness during the test; essentially to see whether “peak-end” rule observed in pain research also occurred with dyspnea. They found that 2 min additional exercise at lower intensity did not affect symptom recall of breathlessness intensity.

Strengths

• This reviewer was initially puzzled by rationale or even necessity of such a trial, but authors laid good foundation & convinced me in their Introduction

• Adequate N, with good statistical power

• Single center, prospective, with good randomization

Weaknesses

• It was not clear why these subjects were referred for exercise testing, but reader later had clue from end of Results section (fitness test for firefighters).

• This limits generalizability since subjects generally had low MRC dyspnea scores – not the type one might encounter in Pulmonary Rehab program. I think they realized this in line 284+

Abstract

No comments. Reads clearly enough.

Intro

Premise of the study well developed.

Methods

• Authors should state why these patients were referred for CPX. Examination of Table shows remarkable absence of morbidity in study population…most were healthy?

• Incl:Excl criteria make sense

• Authors collected much data that was not reported. One item in particular seemed strange (ll. 124-5: how would subjects know a priori what maximal exercise test would feel like?)

• I missed results of questions posed in ll. 148-9. What would their relevance be?

• Frequency of dyspnea & perceived exertion assessments in text (up to lines 140) does not match results shown in Table. Text does not state these symptoms rated during recovery, yet such data appears in Table 2 even for control group (unless this is what authors meant whent they wrote “more often when needed” in line 139?).

• Post-exercise question posed in ll. 153-4 were not asked pre-exercise, were they?

Results

These are reported clearly, Tables complementing text, except Table 4 which only causes confusion. My only question stems from Table 2: how did they obtain Borg CR 10 rating 6.80 (1.53) in controls during recovery, & why did it differ from peak exercise 6.53? This relates to my bullet points in Methods above.

Discussion

In reality, this could be shortened considerably, particularly paragraph on implications. It is here that authors introduce concept of “larger total amount of breathlessness” as if they’re reporting area under curve. In reality, they had intervention group exercise 2 more minutes to cool down & obtained another dyspnea rating, which they did not to in controls (ll. 168-70). Authors state the obvious in ll. 294-5, as this result was inherent in study design.

Minor Points:

1. Line 67: expected or anticipated is better word than “predicted”.

2. NOTE that Figure 2 is not mentioned in text of manuscript; only in Figure legend.

Reviewer #3: The aims of the randomized controlled clinical trial were to (1) test the effect of adding two minutes of low intensity exercise (intervention) on breathlessness, (2) analyze the impact of peak and end exertional breathlessness intensity on breathlessness recall. No significant difference was observed in recalled breathlessness intensity between the arms. However, the intervention arm had significantly lower end breathlessness. The manuscript was written with clarity and a high level of detail.

Minor revisions:

1- Line 162: State the underlying covariance structure used in the mixed effects linear model and the criteria for selecting it.

2- Line 177: Indicate the statistical testing method which achieves 80% power and if the alpha level was one- or two-sided.

3- Provide more precise p-values than “p < 0.05.”

4- Line 229: Typographical error sensitivity analysis results are not located in Table 1.

6. PLOS authors have the option to publish the peer review history of their article (what does this mean?). If published, this will include your full peer review and any attached files.

Reviewer #1: Yes: Olivia Kate Harrison

Reviewer #2: No

Reviewer #3: No

---

## [Author Response · Author response to Decision Letter 0]

29 Jul 2020

The study protocols and aims are also very explicit, and the null finding discussed in a thorough manner.

Thank you.

1. Patient population not described in abstract nor methods – this makes the reader curious as to why they were referred for exercise testing? There must have been other existing conditions that lead to this prescription for an exercise test? This is somewhat covered in the table of patient demographics, so it would be useful to have an in-text reference to this. A short overview in the methods section text would also be helpful.

Thank you for this suggestion. We agree and hade added the following line to the methods section, page 4, paragraph 3: 

“The reason for referral were in most cases suspected chronic coronary syndrome. Other reasons for the test were suspected exercise induced arrhythmias, occupational health screenings and breathlessness.”

2. The question “How are your physical capacity” does not make grammatical sense in English – is there a more appropriate translation (I imagine this question was delivered in Swedish originally)?

Yes, it was translated from Swedish. We agree and have changed this to, page 7, paragraph 2:

“Estimate your physical capacity”

3. Statistical analyses: Were non-normally distributed variables compared using non-parametric tests? It is stated that they are described as a mean ± IQR, but were they also compared between groups in a non-parametric fashion instead of t-tests? It is not clear from the current description in the methods. It would also be helpful to have an additional column in the table of results that depicts which test (with the parametric or non-parametric nature) was employed for each variable.

Thank you, we have compared durations (that were expected to be non-normally distributed) with Wilcoxon rank-sum test instead (table 2). Durations in table 2 were Wilcoxon rank-sum test were used were marked with “*”. We also confirmed normal distribution, of the variables were t-tests were used, with histogram plots. A line was added to reflect this, page 8, paragraph 1:

“Normal distribution was confirmed using histogram plots. Non-normal continuous variables were compared using Wilcoxon Rank-Sum test.”

4. Table 4 is a little confusing. Could this be adjusted to two lines, as I understand the last two lines are part of the same regression analysis, with the column on the left depicting whether the estimates were calculated using separate or simultaneous regressions?

Thank you for this comment. We understand how this could cause confusion and have fused the last two rows in table 4, page 14. 

5. Figure 2 would benefit from error bars on the graph, so that it is possible to visualise the variability in participant responses at each time point. A vertical line at the end of the incremental exercise might also be helpful to see when incremental exercise was terminated. Alternatively (or additionally in a second Figure panel), it could be helpful to see breathlessness scores plotted against exercise intensity - possibly as a percentage of maximum for consistency across participants.

We thank the reviewer for raising these points. Error bars have been added. Importantly, the same exercise protocols were used and the increases in exercise intensity per time unit were similar in both group due to the randomization. Therefore, plotting by exercise intensity such as W or W%max) on the x-axis give similar results and, in our opinion, does not provide anything beyond the data presented in the current graph. The termination time points (and work rates) varied between individuals in both groups and, therefore, it is not feasible to plot these. The figure provides the actual values for different time points throughout the exercise tests for both randomized groups.

6. How were the missing values dealt with? Participants excluded, a correction method used? It would be helpful if this was described.

Due to the standardized test and assessment protocol, there were very few missing data points (as described in Methods page 10). We agree this needs clarification and have added a statement that no data were imputed, on page 12, paragraph 3. The number of missing data points has also been corrected. 

7. It might also be useful to plot and test the correlation between peak and observed breathlessness, to see if this is the reason for no additional variance being explained when both regressors are included in the analysis together.

We think that the reviewer here asks about the correlation between the peak and end breathlessness; that correlation, using linear regression, was 0.62 (please see the scatterplot below, r = 0.68). The correlation was quite high so we agree with the reviewer that this probably at least partly explains the lack of added variance explained. There is a short comment on ‘end breathlessness’ not being independently associated (when adjusting for the peak value) with recall in the discussion section, page 15, paragraph 3 

Reviewer #2: 

Abstract

No comments. Reads clearly enough.

Thank you. 

Intro

Premise of the study well developed.

Thank you. 

Methods

1. Authors should state why these patients were referred for CPX. Examination of Table shows remarkable absence of morbidity in study population…most were healthy?

Thank you, we agree and have added a line on this as also suggested by Reviewer 1, page 4, paragraph 3. The participants were reasonably healthy. Most tests were done because of suspected chronic coronary syndrome. However, participants with a higher pre test probability (meaning more comorbidities) of disease often did other types of tests like myocardial perfusion imaging. 

2. Authors collected much data that was not reported. One item in particular seemed strange (ll. 124-5: how would subjects know a priori what maximal exercise test would feel like?)

We anticipated that participants would be able to estimate how it would feel as most people in Sweden are used to biking. As we did not use this data we have deleted it from the manuscript. 

3. I missed results of questions posed in ll. 148-9. What would their relevance be?

Results are reported in table 3. It was hypothesized that participants who were randomized to the intervention group would anticipate a lower expected (future) breathlessness/exertion (the rationale being that they would recall being less breathless and therefore might have predicted that they would become less breathless in a repeated test) and having a higher physical capacity if a new test were to be done.

4. Frequency of dyspnea & perceived exertion assessments in text (up to lines 140) does not match results shown in Table. Text does not state these symptoms rated during recovery, yet such data appears in Table 2 even for control group (unless this is what authors meant whent they wrote “more often when needed” in line 139?).

Thank you, we have corrected this. Breathlessness, and chest pain as needed, was also inquired about during recovery. Exertion was however not. We have corrected table 2 and added the following line, page 6, paragraph 3:

“breathlessness intensity was self-reported by the participant before the test, every 2 minutes during exercise, at peak exercise, at the end of the intervention period, and at the completion of the test on a Borg CR10 scale”

5. Post-exercise question posed in ll. 153-4 were not asked pre-exercise, were they?

The reviewer is correct, only the question on expected intensity of breathlessness were asked both before and after the test.

Results

These are reported clearly, Tables complementing text, except Table 4 which only causes confusion. My only question stems from Table 2: how did they obtain Borg CR 10 rating 6.80 (1.53) in controls during recovery, & why did it differ from peak exercise 6.53? This relates to my bullet points in Methods above.

Thank you for noticing, this is mostly likely caused by some measurements being taken immediately after cessation of exercise, when the oxygen debt is at its highest with the patient having a further increase of ventilation. In this setting the estimated breathlessness rating can be higher than at peak exercise. The same phenomena likely could be seen in some patients in the intervention group also. We have added a line in the discussion section regarding this page 16, paragraph 1:

 “Peak breathlessness was marginally (non-significantly) higher immediately post exercise in the control group, which was likely caused by the buildup of oxygen debt with the participants likely having further increased ventilation after cessation of exercise.”

Discussion

In reality, this could be shortened considerably, particularly paragraph on implications. It is here that authors introduce concept of “larger total amount of breathlessness” as if they’re reporting area under curve. In reality, they had intervention group exercise 2 more minutes to cool down & obtained another dyspnea rating, which they did not to in controls (ll. 168-70). Authors state the obvious in ll. 294-5, as this result was inherent in study design.

Thank you, we have struck a few lines from the discussion section that we felt were superfluous. We have also changed “larger amount of breathlessness” to “longer duration of breathlessness”. 

Minor Points:

1. Line 67: expected or anticipated is better word than “predicted”.

Thank you, we agree and have changed this on page 3, paragraph 3 and throughout the manuscript.

2. NOTE that Figure 2 is not mentioned in text of manuscript; only in Figure legend.

Thank you for noticing this, we have added the following line, page 10, paragraph 1:

“Figure 2 shows the mean trajectory of breathlessness by study group.”

Reviewer #3: 

The aims of the randomized controlled clinical trial were to (1) test the effect of adding two minutes of low intensity exercise (intervention) on breathlessness, (2) analyze the impact of peak and end exertional breathlessness intensity on breathlessness recall. No significant difference was observed in recalled breathlessness intensity between the arms. However, the intervention arm had significantly lower end breathlessness. The manuscript was written with clarity and a high level of detail.

Minor revisions:

1- Line 162: State the underlying covariance structure used in the mixed effects linear model and the criteria for selecting it.

This was a typo. Mixed effect linear models were used in an earlier draft to create figure 2. We have removed this line as we now use the actual values instead. 

2- Line 177: Indicate the statistical testing method which achieves 80% power and if the alpha level was one- or two-sided.

This has been clarified as suggested, by adding “using Student’s t-test” and specifying that the p-values were two-sided.

3- Provide more precise p-values than “p < 0.05.”

We have added precise p-values, page 10, paragraph 3.

4- Line 229: Typographical error sensitivity analysis results are not located in Table 1.

Thank you, we have corrected this.

---

## [Editor Report · Decision Letter 1]

5 Aug 2020

PONE-D-20-04503R1

Effect of the trajectory of exertional breathlessness on symptom recall and anticipation: a randomized controlled trial

PLOS ONE

Dear Dr. Elmberg,

Thank you for submitting your manuscript to PLOS ONE. After careful consideration, we feel that it has merit but does not fully meet PLOS ONE’s publication criteria as it currently stands. Therefore, we invite you to submit a revised version of the manuscript that addresses the points raised during the review process.

I participated as a reviewer for the initial evaluation of this manuscript, and have now taken on the role of Guest Editor for the consideration of your submission. There are still two areas of concern that need to be addressed before the manuscript can be accepted for publication: 1) The availability of data according to PLOS guidelines, and 2) The response to some of the reviewers’ previous suggestions.

Data availability: PLOS guidelines stipulate that a ‘minimal dataset’ must be shared with a publication (guidelines: https://journals.plos.org/plosone/s/data-availability). This minimal dataset includes the key data values that were inferred upon in the manuscript, such as the underlying values that went into the group mean and regression values reported in Tables 2-4. Ideally data would be placed in a public access repository (strongly recommended), or alternatively the data can be shared within the supplementary material. It is not necessary for authors to submit their entire dataset, for instance any personal or identifying information should not be submitted, and as the sensitive data (such as health information) was used in a purely descriptive sense in this manuscript, there is no need for this data to be made available. Therefore, it should be possible to provide spreadsheets of the exercise data with no subject identification whatsoever, such that participants could never be identified. If the authors can provide reference to documentation from the relevant ethics committee that states making the data publicly available in this form is still not possible, only then would it be considered appropriate for interested parties to initiate contact to request access to the data. However, providing the details of the corresponding author for this purpose is not considered appropriate in the data sharing policies provided by PLOS. Instead it is required that the authors provide contact information for a data access committee, ethics committee, or other institutional body to which data requests may be sent (https://journals.plos.org/plosone/s/data-availability). This ensures a more stable data access point than a single researcher.Comments pertaining to previous reviewer suggestions and response from authors:Figure 2: Regarding a previous point raised by the reviewers, plotting the values as a function of time does not represent the data protocol used in this study. It would be better if the data were plotted as a value that could be standardised across participants, such as a percentage of maximal effort, such that it can be clearly delineated which breathlessness values were from the incremental exercise and which were from the intervention period following the exercise. With individuals stopping at different points in time, plotting breathlessness as a function of time is not informative. Additionally, it would be more helpful to the reader if all error bars were plotted rather than having some omitted as it is currently formatted.Table 4: This is still very confusing for a reader to interpret, as stated in the previous reviewer comments. Perhaps changing the ‘Adjusted for’ column to ‘Regression model’, and describing the rows as either a single regressor or using separate models (row 1), and using multiple regressors or one model (row 2) or something similar might provide more clarity here?It is not currently clear how many of the points within the discussion paragraph on the study implications (beginning on line 337) have direct relevance to the results presented in this paper. In particular, nowhere in the results is the relationship between recalled and experienced breathlessness shown, nor that peak breathlessness is closely associated with future anticipated breathlessness. This paragraph could be removed from the manuscript, as discussed in the previous round of reviewer comments.

We look forward to receiving your revised manuscript.

Kind regards,

Olivia Faull

Academic Editor

PLOS ONE

---

## [Author Response · Author response to Decision Letter 1]

25 Aug 2020

Academic editor

1. Data availability: PLOS guidelines stipulate that a ‘minimal dataset’ must be shared with a publication (guidelines: https://journals.plos.org/plosone/s/data-availability). This minimal dataset includes the key data values that were inferred upon in the manuscript, such as the underlying values that went into the group mean and regression values reported in Tables 2-4. Ideally data would be placed in a public access repository (strongly recommended), or alternatively the data can be shared within the supplementary material. It is not necessary for authors to submit their entire dataset, for instance any personal or identifying information should not be submitted, and as the sensitive data (such as health information) was used in a purely descriptive sense in this manuscript, there is no need for this data to be made available. Therefore, it should be possible to provide spreadsheets of the exercise data with no subject identification whatsoever, such that participants could never be identified. If the authors can provide reference to documentation from the relevant ethics committee that states making the data publicly available in this form is still not possible, only then would it be considered appropriate for interested parties to initiate contact to request access to the data. However, providing the details of the corresponding author for this purpose is not considered appropriate in the data sharing policies provided by PLOS. Instead it is required that the authors provide contact information for a data access committee, ethics committee, or other institutional body to which data requests may be sent (https://journals.plos.org/plosone/s/data-availability). This ensures a more stable data access point than a single researcher.

Thank you. We have anonymized our data and published a minimal dataset at Hardvard dataverse. The dataset contains all necessary data to replicate the study findings. The data availability statement has been updated to reflect this:

“All relevant data are available at the Hardvard dataverse repository at https://doi.org/10.7910/DVN/QMQOCV”

2. Figure 2: Regarding a previous point raised by the reviewers, plotting the values as a function of time does not represent the data protocol used in this study. It would be better if the data were plotted as a value that could be standardised across participants, such as a percentage of maximal effort, such that it can be clearly delineated which breathlessness values were from the incremental exercise and which were from the intervention period following the exercise. With individuals stopping at different points in time, plotting breathlessness as a function of time is not informative. Additionally, it would be more helpful to the reader if all error bars were plotted rather than having some omitted as it is currently formatted.

Thank you we have edited Figure 2 to now show percentage of maximal Watt on the X axis. In retrospect we believe that this much improves the figure and thank the editor for this valuable suggestion. We have also added error bars for all data points. As we nonetheless believe that it could be of some interest to see the breathlessness score relative to the actual time we have included “the old” Figure 2 under supplements as Figure 3. 

3. This is still very confusing for a reader to interpret, as stated in the previous reviewer comments. Perhaps changing the ‘Adjusted for’ column to ‘Regression model’, and describing the rows as either a single regressor or using separate models (row 1), and using multiple regressors or one model (row 2) or something similar might provide more clarity here?

We agree that table 4 required further clarification and have adjusted it as below. We hope that it will now be more understandable. 

4. It is not currently clear how many of the points within the discussion paragraph on the study implications (beginning on line 337) have direct relevance to the results presented in this paper. In particular, nowhere in the results is the relationship between recalled and experienced breathlessness shown, nor that peak breathlessness is closely associated with future anticipated breathlessness. This paragraph could be removed from the manuscript, as discussed in the previous round of reviewer comments.

Thank you. We have reworked some phrases and considerably shortened this paragraph to be more relevant to the study findings. Regarding “experienced breathlessness” it was meant to illustrate that the intervention group, who biked longer compared to the control group, did not have a higher recalled breathlessness. We however agree that this could be confusing, especially considering the study hypothesis and different phrasing and have thus deleted this. 

Results regarding “anticipated future breathlessness” is reported on page 14, paragraph 1:

“Future anticipated breathlessness if a new stress test were to be performed were also more strongly related to peak than end breathlessness although the associations were weaker (r2 = 0.16 vs. 0.11; p = 0.006 and 0.025 respectively). Again, end breathlessness did not increase the variance of recalled breathlessness explained above that for peak breathlessness.“ 

We hope that this is satisfactory as we believe that the findings could potentially have implications on cardiopulmonary exercise programs and that this should be discussed to at least some extent.

---

## [Editor Report · Decision Letter 2]

27 Aug 2020

Effect of the trajectory of exertional breathlessness on symptom recall and anticipation: a randomized controlled trial

PONE-D-20-04503R2

Dear Dr. Elmberg,

We’re pleased to inform you that your manuscript has been judged scientifically suitable for publication and will be formally accepted for publication once it meets all outstanding technical requirements.

Kind regards,

Olivia Faull

Guest Editor

PLOS ONE
---

## [Editor Report · Acceptance letter]

1 Sep 2020

PONE-D-20-04503R2 

Effect of the trajectory of exertional breathlessness on symptom recall and anticipation: a randomized controlled trial 

Dear Dr. Elmberg:

I'm pleased to inform you that your manuscript has been deemed suitable for publication in PLOS ONE. Congratulations! Your manuscript is now with our production department. 

Kind regards, 

on behalf of

Dr. Olivia Faull 

Guest Editor

PLOS ONE